# Abdominal Wall Hernias—State of the Art of Laparoscopic versus Robotic Surgery

**DOI:** 10.3390/jpm14010100

**Published:** 2024-01-16

**Authors:** Pietro Anoldo, Michele Manigrasso, Anna D’Amore, Mario Musella, Giovanni Domenico De Palma, Marco Milone

**Affiliations:** 1Department of Advanced Biomedical Sciences, “Federico II” University of Naples, 80138 Naples, Italy; mario.musella@unina.it; 2Department of Clinical Medicine and Surgery, “Federico II” University of Naples, 80138 Naples, Italy; michele.manigrasso@unina.it (M.M.); anna.damore1993@libero.it (A.D.); giovanni.depalma@unina.it (G.D.D.P.); marco.milone@unina.it (M.M.)

**Keywords:** abdominal wall hernia, laparoscopic surgery, robotic surgery, minimally invasive surgery, inguinal hernia, ventral hernia

## Abstract

Abdominal wall hernia repair, a common surgical procedure, includes various techniques to minimize postoperative complications and enhance outcomes. This review focuses on the comparison between laparoscopic and robotic approaches in treating inguinal and ventral hernias, presenting the ongoing situation of this topic. A systematic search identified relevant studies comparing laparoscopic and robotic approaches for inguinal and ventral hernias. Randomized control trials, retrospective, and prospective studies published after 1 January 2000, were included. Search terms such as hernia, inguinal, ventral, laparoscopy, robotic, and surgery were used. A total of 23 articles were included for analysis. Results indicated similar short-term outcomes for robotic and laparoscopic techniques in inguinal hernia repair, with robotic groups experiencing less postoperative pain. However, longer operative times and higher costs were associated with robotic repair. Robotic ventral hernia repair demonstrated potential benefits, including shorter hospital stay, lower recurrence and lower reoperation rates. While robotic surgery offers advantages such as shorter hospital stays, faster recovery, and less postoperative pain, challenges including costs and training requirements need consideration. The choice between laparoscopic and robotic approaches for abdominal wall hernias should be tailored based on individual surgeon expertise and resource availability, emphasizing a balanced evaluation of benefits and challenges.

## 1. Introduction

Hernia repair stands as a ubiquitous surgical intervention on a global scale, addressing a prevalent health concern that transcends geographic boundaries. Among the diverse types of hernias, inguinal hernias emerge as the predominant subtype, constituting a substantial portion—75%—of all abdominal wall hernias. The statistical landscape unveils a gender-specific prevalence, with approximately 25% of men and a comparatively meager 2% of women encountering inguinal hernias over the course of their lifetimes [1]. Meanwhile, ventral incisional hernias, arising as a frequent complication of abdominal surgeries, demonstrate an incidence rate fluctuating between 2% and 20%, displaying significant variability across different medical series [2].

The evolution of hernia repair techniques has been dynamic, encompassing a spectrum from traditional open procedures employing primary sutures to contemporary approaches that prioritize minimizing postoperative complications and enhancing overall outcomes, including recovery and recurrence rates. Within this paradigm, laparoscopic procedures have emerged as a transformative avenue, demonstrating noteworthy advantages over their conventional counterparts. Studies indicate that laparoscopic interventions are associated with a decreased infection rate, shorter hospital stays, and diminished persistent pain, collectively contributing to an improved patient experience [3]. Amid the landscape of minimally invasive approaches, the robotic methodology has garnered considerable attention as a promising alternative to laparoscopy in hernia repair. Noteworthy differentiators include the feasibility of suturing, as opposed to tack fixation, and enhanced surgeon ergonomics. The robotic approach introduces a paradigm shift by combining precision with adaptability, potentially reshaping the landscape of hernia repair.

This literature review aims to cast a comprehensive gaze upon the current state of minimally invasive treatments for inguinal and ventral hernias, drawing a comparative analysis between the established laparoscopic approach and the emerging robotic alternative. By delving into the nuances of these methodologies, we seek to unravel the intricate web of advantages and potential drawbacks, thereby providing a nuanced understanding of the evolving landscape of hernia repair. As medical science continues to progress, the insights gleaned from this exploration may pave the way for refined surgical strategies, optimizing patient outcomes and steering the trajectory of hernia repair into the future.

## 2. Materials and Methods

In our quest to comprehensively elucidate the landscape of minimally invasive surgery for abdominal wall hernias, particularly focusing on the comparative efficacy of laparoscopic and robotic approaches, a systematic and rigorous approach was employed. To ensure methodological robustness, we conducted an extensive literature search across multiple electronic databases, including PubMed, Web of Science, Scopus, and EMBASE, adhering meticulously to the Preferred Reporting Items for Systematic Reviews and Meta-Analyses (PRISMA) guidelines [4]. The utilization of PRISMA guidelines ensured transparency and reproducibility in our search strategy and subsequent analyses.

The search criteria were meticulously defined to capture the most relevant studies. We restricted our inclusion criteria to articles published after 1 January 2000, aiming for a contemporary and relevant synthesis of evidence. To ensure a comprehensive representation of the available literature, we considered a diverse array of study designs, encompassing randomized controlled trials, retrospective, and prospective studies. Our deliberate inclusion of studies across varied designs aimed at capturing a nuanced understanding of the comparative effectiveness of laparoscopic and robotic approaches in the treatment of both inguinal and ventral hernias.

To refine our search strategy, we utilized a combination of key terms, namely “hernia”, “inguinal”, “ventral”, “laparoscopy”, “robotic”, and “surgery”. These terms were systematically combined to maximize the retrieval of pertinent literature, ensuring that our analysis would be both comprehensive and thorough.

In alignment with our commitment to methodological rigor, we exclusively considered English-written articles. This deliberate choice was made to uphold consistency in language and to facilitate a clear and unambiguous review process. By adhering to stringent inclusion criteria, we aimed to provide a focused and insightful overview of the current state of the art in minimally invasive surgery for abdominal wall hernias, with a specific emphasis on the comparative merits of laparoscopic and robotic interventions in the treatment of inguinal and ventral hernias. The outcomes mainly taken into consideration when comparing the two techniques were length of hospital stay, postoperative pain, complications including conversions and recurrence.

## 3. Results

The initial systematic search yielded a substantial pool of 4086 articles, reflecting the breadth of research in the field of minimally invasive surgery for abdominal wall hernias. Following a meticulous curation process, 2858 duplicate articles were expunged, streamlining the dataset and ensuring the integrity of the subsequent analyses. The winnowed-down set of 1228 unique articles underwent a thorough evaluation by two independent authors, who systematically appraised each article’s relevance based on predetermined criteria.

Articles that did not directly compare the laparoscopic and robotic approaches for the treatment of inguinal or ventral hernias were systematically excluded, as were those not written in English. This stringent selection process aimed to distill the literature down to the most pertinent and directly comparable studies. Furthermore, reviews were intentionally omitted to maintain a focus on primary research and to ensure the inclusion of original data.

Following this rigorous evaluation, a total of 23 articles emerged as meeting the stringent inclusion criteria. Among these, 10 articles specifically compared laparoscopic and robotic approaches in the context of ventral hernia treatment, comprising four randomized controlled trials and six retrospective studies. Additionally, 12 articles explored the comparative efficacy of laparoscopic and robotic approaches in inguinal hernia treatment, with the distribution including one randomized controlled trial, nine retrospective studies, and two prospective studies. Notably, one retrospective study undertook a comparative analysis of both ventral and inguinal hernia treatments using laparoscopic and robotic approaches (Figure 1).

For the convenience of readers, the articles included in this comprehensive review have been meticulously reported and organized chronologically in Table 1, Table 2 and Table 3. This systematic arrangement allows for a structured and accessible overview of the evolving landscape of research on the comparison between laparoscopic and robotic approaches in the treatment of inguinal and ventral hernias.

## 4. Discussion

Robotic surgery has emerged as a groundbreaking technology in several fields of general surgery including gastroesophageal, colorectal and hepatobiliopancreatic procedures, even in emergency setting [27,28,29,30]. The application of robotic platforms in abdominal wall hernia repair has been a topic of considerable discussion. Traditional open surgery and laparoscopic techniques have been the standard approaches for hernia repair, but robotic surgery introduces a new dimension to this field. The robotic instruments can mimic the movements of the human hand with enhanced precision, allowing for intricate maneuvers in the confined space of the abdominal cavity. This level of precision is particularly crucial in hernia repair, where careful manipulation of tissues and accurate placement of mesh are essential for a successful outcome, especially if the preparation of the musculofascial flaps requires the need to act on opposite abdominal quadrants [31]. Despite the advantages that robotic surgery hopes to bring to the treatment of these pathologies, clinical studies comparing the outcomes of laparoscopic and robotic hernia repair have shown comparable results in terms of postoperative outcomes, complication rates and recurrence. Furthermore, the economic sustainability remains an open issue about robotic surgery [32].

Regarding the minimally invasive treatment of inguinal hernias, the studies examined by this review generally highlighted similar short-term outcomes when comparing robotic and laparoscopic techniques. In five non-RCT articles, robotic groups were associated with less postoperative pain assessed with visual analogue scale [5,7,13], verbal rating scale [11] and 1–10 scale [16]. The robotic groups were associated with longer operative time in five articles, but this characteristic must be related to expertise in robotic surgery; Ayuso et al. found that operative time was longer in the robotic group, but when evaluating robotic procedures at the beginning of the study versus the end of the study, there was a 50-min decrease in operative time [9]. The 2-year outcomes of the RIVAL trial showed that laparoscopic and robotic inguinal hernia repairs have similar long-term outcomes when performed by surgeons with experience in minimally invasive inguinal hernia repairs; their long-term results indicate that robotic repairs were performed competently, showing no evidence of a learning curve effect. The surgeons, experienced in minimally invasive surgery groin anatomy and technique, demonstrated similar outcomes for both laparoscopic and robotic repairs. The robotic technique of mesh fixation, involving sewing rather than tacking, did not result in significant differences in postoperative pain at 1 and 2 years compared to laparoscopic repair. This challenges the notion that long-term pain after minimally invasive inguinal hernia repair is strongly influenced by mesh fixation methods [12]. Abdelmoaty et al. evaluated the cost-effectiveness of robotic-assisted surgery, particularly in inguinal hernia repair, compared to laparoscopic approaches; their findings reveal that robotic-assisted inguinal hernia repair is associated with significantly higher total costs compared to laparoscopic approaches, with an average difference of $2200 per case. The major contributors to this cost difference are identified as the medical device costs and personnel costs, with longer operative times for the robotic approach indirectly impacting personnel costs. [15].

Similar outcomes were also found in studies comparing robotic and laparoscopic procedures in the treatment of ventral hernias. Robotic groups, compared to laparoscopic ones, are characterized by longer operating times, but also by a shorter length of hospital stay [18,23,24,26]. The findings of the trial from Costa et al. suggest that both laparoscopy and robotics exhibit comparable efficacy in terms of short- and long-term outcomes; the study emphasizes the potential benefits of robotic-assisted surgery, particularly in ventral hernias, where the degrees of freedom provided by robotic technology significantly impact dexterity. The articulating instruments in robotic surgery overcome angle restrictions associated with laparoscopy, potentially contributing to a higher successful closure rate of hernia defects. Despite a longer operating room time for robotic ventral incisional hernia repair, the study notes that this did not influence the rate of complications, which were similar between laparoscopic and robotic groups. Both approaches showed equivalence in terms of postoperative morbidity. In the long-term follow-up, no significant differences were noted between laparoscopic and robotic hernia repairs in terms of hernia recurrence, abdominal wall strength (evaluated by Kendall’s test) and quality of life at the 24-month mark. The article acknowledges the limitations of quality of life assessment, noting that deterioration may be related to cancer aftereffects rather than the hernia itself [20].

In some of the articles included in the review, the robotic procedure appeared safer. Bowel injuries during minimally invasive ventral hernia repair led to elevated rates of postoperative complications, including wound morbidity, enterocutaneous fistula, reoperation, septic shock, myocardial infarction, acute renal failure, and intubation for respiratory failure. The study by Thomas et al. investigates the impact of minimally invasive approaches, specifically laparoscopic and robotic, on bowel injury during ventral hernia repair using a national registry: with over 10,000 patients included, the findings reveal that laparoscopic ventral hernia repair is associated with a higher risk of bowel injury compared to robotic repair. Notably, the study is one of the first to demonstrate an increase in reoperations for missed enterotomies during laparoscopic ventral hernia repair compared to robotic repair. These results suggest that robotic ventral hernia repair is less likely to result in bowel injury than laparoscopic repair, contributing valuable data to the complex debate around the choice between these two approaches [21]. The article by LaPinska et al. delves into the ongoing debate over the ideal surgical management for ventral hernias, particularly comparing laparoscopic ventral hernia repair and robotic ventral hernia repair: utilizing real-world evidence from the Americas Hernia Society Quality Collaborative (AHSQC), the study represents the largest matched-case series analysis of these approaches without myofascial release. Their study reveals that patients undergoing robotic ventral hernia repair experienced a shorter hospital length of stay, despite comparable rates of intraoperative complications. Notably, the robotic group had fewer conversions to laparotomy, possibly attributed to robotic 3D visualization and wristed technology facilitating precise assessment of anatomy and adhesion lysis. While overall complications did not significantly differ between robotic repair and laparoscopic repair, patients in the robotic cohort required less treatment for wound complications. Their article suggests that the wristed facilitation of fascial closure with the robotic-assisted approach might contribute to better surgical site outcomes, although further investigation is needed [23]. The study by Dhanani et al. represents one of the initial randomized controlled trials providing long-term results of robotic ventral hernia repair. In the hands of high-volume experts, the research suggests that robotic ventral hernia repair is safe and effective compared to laparoscopic repair, with potential benefits including fewer reoperations observed at the 2-year mark. Notably, the study indicates statistically fewer reoperations with robotic surgery at the 2-year follow-up, accompanied by a lower percentage of hernia recurrences. However, the study acknowledges limitations, as it was initially conducted to detect differences in the length of hospital stay at 90 days, and the findings regarding reoperations remain hypothesis-generating. They concluded that, at the 2-year follow-up, robotic ventral hernia repair appears to be safe and effective compared to laparoscopic ventral hernia repair, potentially with benefits such as decreased hernia recurrence and reoperation, although a larger-scale study with the specific goal of assessing recurrence and reoperation outcomes should be conducted to draw more definitive conclusions [17].

Zayan et al. present a comprehensive analysis of patient demographics and outcomes in the context of laparoscopic and robotic approaches for inguinal and ventral hernia repairs.

Regarding inguinal hernia repairs, the robotic group had a higher proportion of bilateral repairs, which, counterintuitively, goes against the hypothesis, suggesting that more extensive procedures did not necessarily lead to increased complications or pain. Patient-reported outcomes, assessed through the Carolinas Comfort Scale (CCS), showed no significant differences between laparoscopic and robotic inguinal hernia repairs at various postoperative time points. The authors emphasize that the statistical difference in follow-up time is not deemed clinically significant. Similar patterns were observed in ventral hernia repairs, with no significant disparities in patient-reported outcomes between laparoscopic and robotic approaches. The robotic ventral hernia repair exhibited a lower baseline CCS but a higher interval CCS, which the authors attribute to potential selection bias, and acknowledge that both robotic cases and laparoscopic cases had comparable CCS scores at 1 year [3].

The adoption of a robotic approach may be due to the ease of the surgical gesture that it allows. Okamoto et al. found that dissection time for a medial-type hernia in the robotic group was marginally shorter than that in the laparoscopic group; they stated that this result may arise from the advantages of robotic surgery, such as three-dimensional magnified view, wristed instruments and tremor filtration [6]. Warren et al. concluded that the robotic approach enables true abdominal wall reconstruction thanks to its extensive dissection and myofascial release; the authors argue that the ergonomics of the robotic platform enhance the dissection of the difficult anterior abdominal wall, and intracorporeal suturing is significantly improved. Robotic ventral hernia repair is considered ideal for cases requiring complex adhesiolysis, extensive musculofascial dissection and large mesh prosthesis placement; furthermore, the robotic approach is associated with a larger mesh size, potentially influencing recurrence rates [26].

Summing up, this review pointed out how robotic surgery offers the advantage of shorter hospital stays, faster recovery times and less pain for patients. The minimally invasive nature of the procedure, facilitated by small incisions and reduced tissue trauma, contributes to less postoperative pain and a quicker return to normal activities. No less important is the ergonomic issue: robotic systems come with intuitive control interfaces, where surgeons can manipulate the robotic arms with hand and foot controls. This design minimizes physical strain on the surgeon by allowing more natural and comfortable movements, reducing fatigue during long procedures. Surgeons can operate while comfortably seated at a console, with ergonomically optimized positioning of controls. This helps minimize musculoskeletal stress and fatigue, promoting a more comfortable and stress-free operating experience. Robotic experts could benefit the most from the ergonomic advantages in robotic surgery [33]. Moreover, robotic surgery systems often have a shorter learning curve compared to traditional laparoscopic techniques due to their intuitive controls, enhanced visualization, and simulator training options. This means that surgeons may become proficient more quickly, leading to more consistent and reliable outcomes over time. However, individual surgeon experience, prior laparoscopic skills and the specific surgical procedure can influence the learning curve for both technologies. Robotic inguinal hernia repair allows minimally invasive surgeons the buildup of a short, safe and efficacious robotic learning experience for future more complex robotic surgeries [34]. However, it is essential to acknowledge that the adoption of robotic surgery for abdominal wall hernias is not without challenges. The cost of the robotic system, training requirements for surgeons and the need for specialized personnel are factors that may limit its widespread implementation.

Operative time is another critical metric in assessing the efficiency and effectiveness of surgical procedures, and this holds true in the realm of robotic surgery. As robotic technology continues to advance, understanding the factors influencing operative time becomes paramount. Surgeon proficiency and familiarity with robotic systems significantly impact operative time. Experienced surgeons often exhibit shorter operative times, emphasizing the importance of ongoing training and skill development. Patient-specific factors such as anatomical variations and comorbidities can influence operative time. Understanding and accounting for these variables is crucial for preoperative planning and resource allocation. Ongoing advancements in robotic technology contribute to reduced operative times by enhancing system capabilities, improving instrumentation, and providing more intuitive interfaces for surgeons. Furthermore, effective communication and collaboration within the surgical team play a crucial role in minimizing delays and optimizing workflow during robotic procedures, ultimately influencing operative time. Furthermore, some studies have suggested that the clinical outcomes of robotic hernia repair may not significantly differ from those of laparoscopic techniques, raising questions about the cost-effectiveness of robotic surgery in this context. As the field continues to evolve, a balanced consideration of both the benefits and challenges will be crucial in determining the optimal place of robotic surgery in the broader landscape of hernia management. Long-term studies assessing factors such as chronic pain and quality of life are still needed to establish the superiority of one approach over the other definitively.

## 5. Conclusions

In conclusion, the management of abdominal wall hernias involves a careful consideration of the specific advantages and limitations of each technique. The choice of the most suitable approach should be based on individual surgeon expertise and resource availability and should be tailored for each patient. This review can be considered a valuable tool for summarizing existing knowledge and identifying trends in minimally invasive treatments of abdominal wall hernias. However, some limitations should be mentioned. Not all studies are of equal quality; comparing and synthesizing findings from studies with different research approaches can be challenging. Studies on the same topic may use different definitions, measures, or methodologies, making it difficult to compare and synthesize findings in a meaningful way. Large-scale randomized trials with long follow-ups will help shed light on the issue.

## 6. Future Directions

The future perspectives of robotic surgery in the treatment of abdominal wall hernias hold promise for advancements in surgical techniques, patient outcomes, and overall healthcare efficiency. Future robotic systems are likely to incorporate improved precision and dexterity, allowing surgeons to perform intricate maneuvers with greater ease. This can be particularly beneficial in hernia repair procedures where precise tissue manipulation is crucial. Integration of AI into robotic platforms may enable real-time data analysis, assisting surgeons in decision-making during hernia repair surgeries. AI algorithms could provide insights into optimal mesh placement, identify anatomical variations, and assist in predicting postoperative outcomes. Future robotic systems may integrate haptic feedback mechanisms to provide surgeons with a sense of touch during procedures. This can enhance the surgeon’s ability to differentiate tissues, leading to more refined and delicate maneuvers during abdominal wall hernia repairs. The expansion of remote or telesurgery capabilities in robotic systems could potentially permit to surgeon to perform hernia repairs from a distance, providing expertise and access to specialized care in regions with limited surgical resources. Three-dimensional printing and personalized medicine may play a role in creating customized mesh implants for abdominal wall hernia repairs; tailoring the mesh to the patient’s specific anatomy could enhance the long-term success of the procedure and reduce the risk of complications. Moreover, continued accumulation of clinical data from robotic hernia repair procedures will enable comprehensive outcome analysis; this can contribute to refining surgical techniques, identifying best practices, and establishing evidence-based guidelines for optimal patient outcomes. Future perspectives may involve a more collaborative approach, with surgeons working closely with engineers, data scientists, and other healthcare professionals to drive innovations in robotic surgery for abdominal wall hernias. This multidisciplinary collaboration could lead to holistic advancements in technology and patient care.

## Figures and Tables

**Figure 1 jpm-14-00100-f001:**
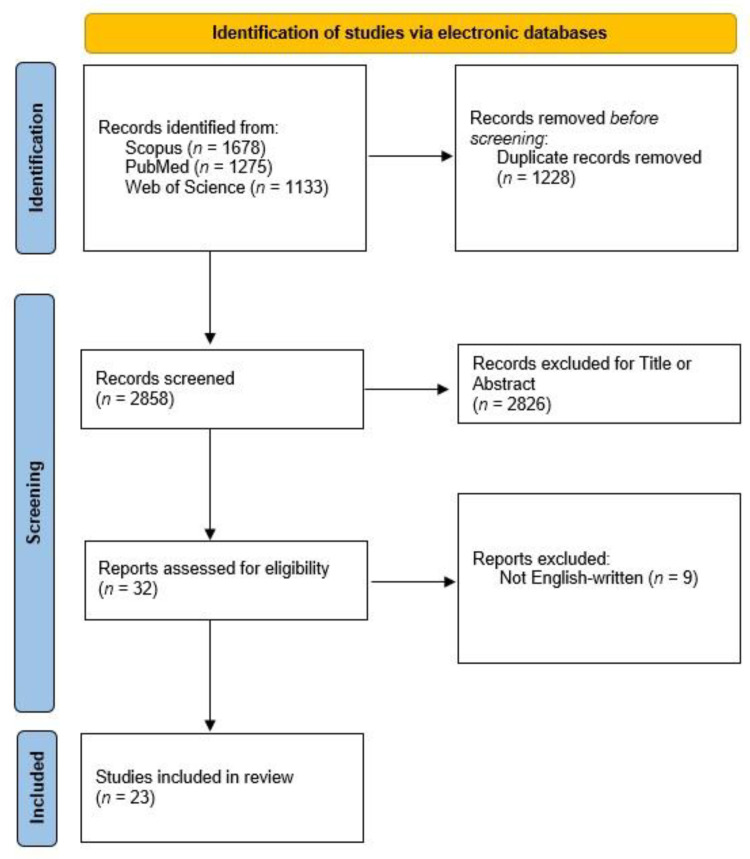
Studies’ exclusion according to PRISMA. PRISMA, preferred reporting items for systematic reviews and meta-analyses.

**Table 1 jpm-14-00100-t001:** Articles on inguinal hernia included.

Author	Title	Year	Journal	Type	Patients	Lap	Rob
Choi et al. [5]	Initial Experience of Robot-Assisted Transabdominal Preperitoneal (TAPP) Inguinal Hernia Repair by a Single Surgeon in South Korea	2023	*Medicina*	Retrospective	100	50	50
Okamoto et al. [6]	Comparison of short-term outcomes of robotic and laparoscopic transabdominal peritoneal repair for unilateral inguinal hernia: a propensity-score matched analysis	2023	*Hernia*	Retrospectve	160	80	80
Vitiello et al. [7]	Minimally Invasive Repair of Recurrent Inguinal Hernia: Multi-Institutional Retrospective Comparison of Robotic Versus Laparoscopic Surgery	2023	*Journal of Laparoendoscopic & Advanced Surgical Techniques*	Retrospective	48	25	23
Peltrini et al. [8]	Robotic versus laparoscopic transabdominal preperitoneal (TAPP) approaches to bilateral hernia repair: a multicenter retrospective study using propensity score matching analysis	2022	*Surgical Endoscopy*	Retrospective	120	80	40
Ayuso et al. [9]	Laparoscopic versus robotic inguinal hernia repair: a single-center case-matched study	2022	*Surgical Endoscopy*	Prospective	282	141	141
Kudsi et al. [10]	Comparison of perioperative and mid-term outcomes between laparoscopic and robotic inguinal hernia repair	2022	*Surgical Endoscopy*	Retrospective	1153	606	547
Gerdes et al. [11]	Results of robotic TAPP and conventional laparoscopic TAPP in an outpatient setting: a cohort study in Switzerland	2022	*Langenbeck’s Archives of Surgery*	Prospective	58	29	29
Miller et al. [12]	Laparoscopic versus robotic inguinal hernia repair: 1- and 2-year outcomes from the RIVAL trial	2022	*Surgical endoscopy*	RCT	102	54	48
Aghayeva et al. [13]	Laparoscopic totally extraperitoneal vs. robotic transabdominal preperitoneal inguinal hernia repair: Assessment of short- and long-term outcomes	2020	*The International Journal of Medical Robotics and Computer Assisted Surgery*	Retrospective	86	43	43
Khoraki et al. [14]	Perioperative outcomes and cost of robotic-assisted versus laparoscopic inguinal hernia repair	2019	*Surgical Endoscopy*	Retrospective	183	138	45
Abdelmoaty et al. [15]	Robotic-assisted versus laparoscopic unilateral inguinal hernia repair: a comprehensive cost analysis	2018	*Surgical Endoscopy*	Retrospective	2405	1671	734
Waite et al. [16]	Comparison of robotic versus laparoscopic transabdominal preperitoneal (TAPP) inguinal hernia repair	2016	*Journal of Robotic Surgery*	Retrospective	63	24	39

**Table 2 jpm-14-00100-t002:** Articles on ventral hernia included.

Author	Title	Year	Journal	Type	Patients	Lap	Rob
Dhanani et al. [17]	Robotic Versus Laparoscopic Ventral Hernia Repair	2023	*Annals of Surgery*	RCT	124	59	65
Christoffersen et al. [18]	Less postoperative pain and shorter length of stay after robot-assisted retrorectus hernia repair (rRetrorectus) compared with laparoscopic intraperitoneal onlay mesh repair (IPOM) for small or medium-sized ventral hernias	2022	*Surgical Endoscopy*	Retrospective	59	32	27
Petro et al. [19]	Robotic vs. Laparoscopic Ventral Hernia Repair with Intraperitoneal Mesh: 1-Year Exploratory Outcomes of the PROVE-IT Randomized Clinical Trial	2022	*Journal of the American College of Surgeons*	RCT	71	33	38
Costa et al. [20]	Robotic-assisted compared with laparoscopic incisional hernia repair following oncologic surgery: short- and long-term outcomes of a randomized controlled trial	2022	*Journal of Robotic Surgery*	RCT	37	19	18
Thomas et al. [21]	Comparing rates of bowel injury for laparoscopic and robotic ventral hernia repair: a retrospective analysis of the abdominal core health quality collaborative	2022	*Hernia*	Retrospective	10,660	4116	6544
Olavarria et al. [22]	Robotic versus laparoscopic ventral hernia repair: multicenter, blinded randomized controlled trial	2020	*BMJ*	RCT	124	59	65
LaPinska et al. [23]	Robotic-assisted and laparoscopic hernia repair: real-world evidence from the Americas Hernia Society Quality Collaborative (AHSQC)	2020	*Surgical Endoscopy*	Retrospective	1230	615	615
Prabhu et al. [24]	Laparoscopic vs Robotic Intraperitoneal Mesh Repair for Incisional Hernia: An Americas Hernia Society Quality Collaborative Analysis	2017	*Journal of the American College of Surgeons*	Retrospective	631	454	177
Chen et al. [25]	Outcomes of robot-assisted versus laparoscopic repair of small-sized ventral hernias	2016	*Surgical Endoscopy*	Retrospective	33	39	72
Warren et al. [26]	Standard laparoscopic versus robotic retromuscular ventral hernia repair	2016	*Surgical Endoscopy*	Retrospective	156	103	53

**Table 3 jpm-14-00100-t003:** Articles on both ventral and inguinal hernia included.

Author	Title	Year	Journal	Type	Lap Inguinal	Rob Inguinal	Lap Ventral	Rob Ventral
Zayan et al. [3]	A direct comparison of robotic and laparoscopic hernia repair: patient-reported outcomes and cost analysis	2019	*Hernia*	Retrospective	68	37	33	16

## Data Availability

The data presented in this study are available on request from the corresponding author.

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
