# Peer review of "Abdominal Wall Hernias—State of the Art of Laparoscopic versus Robotic Surgery"

_jpm, 2024, doi:10.3390/jpm14010100_

Round 1

Reviewer 1 Report

Comments and Suggestions for Authors

Very good article both from the point of view of the topic and from the point of view of presentation.

I would have liked that in the material and method part there was not only the presentation of the articles and the analyzed studies, but also some additional comparison elements.

Author Response

Thank you  for taking the time to review this manuscript. 

Comment: "I would have liked that in the material and method part there was not only the presentation of the articles and the analyzed studies, but also some additional comparison elements."

Response: Good advice. I will update the methods section by adding some comparative elements.

Reviewer 2 Report

Comments and Suggestions for Authors

A pertinent study but the authors failed to discuss the real advantage of robotic over laparoscopic surgery being the better ergonomics and stress- free advantages of robotic surgery. The disadvantage is the value of robotic surgery i.e.cost-effectiveness of robotic surgery which is lower than laparoscopic surgery.

The study was a systematic review of 23 articles that  compared laparoscopic with robotic-assisted approaches in the management of abdominal wall (ventral and inguinal) hernias

The topic is original. It discussed the advantages and disadvantages of these procedures with regard to surgical outcome, postoperative pain , length of hospital stay and cost. . However, it did not add to the  literature the very important aspect of the relative values of these approaches with regard to the cost-effectiveness. The authors failed to emphasise that ironically the learning curve in robotic-assisted surgery may be less steeper than of laparoscopic surgery. The ergonometrics of robotic surgery  with less fatigue gives it the tremendous advantage over laparoscopic surgery. These will render some selection bias in the conclusions and a limitation of the study.

It is an important comparative study on the 2 approaches using a large cohort of patients which gave statistically significant results on the respective surgical outcome, but for the problems of selection bias. A double- blinded study which is difficult in these scenarios would limit the bias.

The methodology should have included  a specific assessment of the value ( cost-effectiveness) using the health economics assessment principles will give more value to the research as both approaches have almost similar surgical outcome.

The conclusions are consistent with the evidence and arguments presented. The authors should discuss the limitations of the study.

Author Response

Thank you for the comment. I find your opinions excellent food for thought. 

Certainly one of the undisputed advantages of robotics is ergonomics and less stress compared to laparoscopy. Unfortunately, there are no quantitative studies in the literature on surgeon stress in the context of abdominal wall hernias, and therefore I was unable to carry out an in-depth analysis on the topic. The same goes for the learning curve, which is known to be less steep in robotic surgery. I will add these topics to the discussion anyway.

Regarding the cost-effectiveness issue, there is not enough information in the literature and specific studies should be conducted on the topic. I have focused my review mainly on a clinical point of view.

I will include a discussion of the limitations of the study in the conclusions.

Reviewer 3 Report

Comments and Suggestions for Authors

I have carefully read the article entitled "Abdominal wall hernias – state of art of laparoscopic versus robotic surgery" by the authors Anoldo P et al in order to evaluate its publication in JPM.

I believe that this is a very up-to-date and current topic, of great interest to the reader since it provides a review of the literature on abdominal wall repair comparing minimally invasive approaches: robotic Vs laparoscopic.

Although I believe the article may be of interest for publication, there are some aspects that would improve, in my opinion, its quality.

First, it is not clear to me whether the authors' goal is to conduct a systematic review, meta-analysis, or literature review. It strikes me that a qualitative review of the articles was carried out but not a quantitative one, which I believe would give greater quality to the manuscript.

On the other hand, I think that mixing two pathologies as different as inguinal and ventral hernias diversifies the topic so much that I think it would be more useful to focus on one of them. Likewise, it does not specify the term "ventral hernia" which includes (umbilical, midline, incisional?).

On line 189, it talks about measuring "abdominal wall strength" but I think it is a difficult concept to quantify. How does the author do it?

Comments on the Quality of English Language

No comments

Author Response

Thank you for the time you took to review the work.

Comment: First, it is not clear to me whether the authors' goal is to conduct a systematic review, meta-analysis, or literature review. It strikes me that a qualitative review of the articles was carried out but not a quantitative one, which I believe would give greater quality to the manuscript.

Response: In the literature there are not many works with great scientific power, so I preferred to do a qualitative analysis. It is a snapshot of the state of the art of this topic that can be an inspiration for conducting comparative, randomized work with adequate sample sizes.

Comment: On the other hand, I think that mixing two pathologies as different as inguinal and ventral hernias diversifies the topic so much that I think it would be more useful to focus on one of them. Likewise, it does not specify the term "ventral hernia" which includes (umbilical, midline, incisional?).

Response: It is an issue that I took into consideration when I decided to carry out this literature review. I chose to analyze the literature on 2 different pathologies to broaden the discussion of the comparison of laparoscopy versus robotics on a comprehensive topic, abdominal wall hernias. The objectives that want to be achieved in this type of surgery are the same (postoperative recovery, pain control, complications and recurrences). I analyzed them separately but in the same article to have a state of the art on abdominal hernia pathology as a whole.

Furthermore, in some works the type of ventral hernia is not specified, so I have considered them in the generic definition of "ventral hernia". If I had divided them by type, when specified, it would have been a scattered analysis and based on a few works.

Your comment is more than appreciable and is a starting point for writing even more sector-specific works.

Comment: On line 189, it talks about measuring "abdominal wall strength" but I think it is a difficult concept to quantify. How does the author do it?

Response: Costa et al. evaluated abdominal wall strength by Kendall’s test. This test requires the patient to keep the abdominal muscles contraction as the legs and trunk are lowered in the supine position. I will specify it in the discussion.

Icona di Verificata con community         Icona di Verificata con community        

Round 2

Reviewer 3 Report

Comments and Suggestions for Authors

Thank you for considering my suggestions